# An Ex Vivo Electroretinographic Apparatus for the mL-Scale Testing of Drugs to One Day and Beyond

**DOI:** 10.3390/ijms241411346

**Published:** 2023-07-12

**Authors:** Lorenzo Cangiano, Sabrina Asteriti

**Affiliations:** 1Department of Translational Research, University of Pisa, 56123 Pisa, Italy; 2Department of Neurosciences, Biomedicine and Movement Sciences, University of Verona, 37134 Verona, Italy

**Keywords:** retina, ERG, mouse, ex vivo, incubation, pharmacology, drug, therapy

## Abstract

When screening new drugs to treat retinal diseases, ex vivo electroretinography (ERG) potentially combines the experimental throughput of its traditional in vivo counterpart, with greater mechanistic insight and reproducible delivery. To date, this technique was used in experiments with open loop superfusion and lasting up to a few hours. Here, we present a compact apparatus that provides continuous and simultaneous recordings of the scotopic a-waves from four mouse retinas for much longer durations. Crucially, each retina can be incubated at 37 °C in only 2 mL of static medium, enabling the testing of very expensive drugs or nano devices. Light sensitivity and response kinetics of these preparations remain in the physiological range throughout incubation, displaying only very slow drifts. As an example application, we showed that barium, a potassium channel blocker used to abolish the glial component of the ERG, displayed no overt side effects on photoreceptors over several hours. In another example, we fully regenerated a partially bleached retina using a minimal quantity of 9-cis-retinal. Finally, we demonstrated that including antibiotic in the incubation medium extends physiological light responses to over one day. This system represents a necessary stepping stone towards the goal of combining ERG recordings with organotypically cultured retinas.

## 1. Introduction

Ex vivo electroretinographic (ERG) recordings have some key advantages over their in vivo counterpart when in depth pharmacology must be conducted on the retina. For instance, the photoreceptor a-wave (or ‘fast PIII’ signal component) can be isolated by removing with glutamatergic agonists/antagonists the b-wave mediated by bipolar cells, and with barium the ‘slow PIII’ wave mediated by Müller glia (reviewed in [1]). Thus, rod and cone phototransduction can be efficiently monitored and reliable estimates obtained of light sensitivity and response kinetics [2]. Furthermore, the ex vivo configuration allows the accurate and reproducible administration of drugs, receptor or ion channel blockers, modulators, small molecules as well as full proteins, without the hindrance of the ocular or blood–brain barriers [2,3,4].

Several different configurations for mammalian ex vivo ERG were developed, each addressing specific experimental requirements such as dual retina recordings [3], b-wave stability [5,6] or affordability by conversion of an existing patch clamp setup [7]. The standard practice in these recordings was to continuously superfuse the retina with fresh extracellular medium in an open loop configuration, at a rate of a 1–5 mL/min [2,6,7,8,9,10,11,12,13,14,15,16]. While this guarantees a stable medium composition throughout the experiment, the overall volumes of liquid at play mean that very expensive drug candidates cannot be reasonably tested for any length of time. Additionally, accurate temperature control and mechanical stability of the tissues can be problematic. To address these limitations, the superfusion must be slowed dramatically or, ideally, dispensed with altogether. Albanna and colleagues [5] found that flow rates of 0.2 mL/min were compatible with recordings lasting reliably for several hours, albeit at an incubation temperature of only 27.5 °C. However, even such a low rate can be unaffordable when testing newly synthesized therapeutic molecules (e.g., [17]) or nano-devices, an issue that will only worsen as (in the future) ex vivo ERG recordings achieve longer and longer durations. For instance, a single 24 h experiment would require 288 mL of drug solution at a pharmacologically meaningful concentration.

The use of brisk superfusion in ex vivo ERG studies contrasts with a recent report demonstrating the extraordinary resilience of the a-wave, which was detectable even in mouse and human retinas extracted several hours post-mortem [18]. Given that retina organotypic cultures survive for days or weeks with only occasional changes of the culture medium [19], we wondered whether, by borrowing features from the different techniques, we could succeed in chronically recording a-waves from retinas incubated without superfusion. By ‘chronically’, we mean being able to continuously monitor light responses beyond the few hours mentioned in previous mammalian studies at body temperature [3,9,14,15], studies that in all cases employed active superfusion. This would enable more demanding tests such as characterizing slow drug release from nano-devices or, if several days could be reached, even screen the effects of therapeutic molecules on fast-progressing models of inherited retinal degeneration [20]. Our main other design requirements were: (i) volume of the bathing medium on the mL scale; (ii) retinas fully immersed to ensure a good exposure to test solutions; (iii) test drugs easy to add during an experiment without major environmental perturbations; (iv) multiple pairs of retinas recorded concurrently. We opted for using Ames’ medium, as it includes physiological bicarbonate-buffering and was originally designed for the long-term maintenance of the mammalian retina [21]. Ames’ was favorably vetted in several ex vivo ERG studies [3,5,16,22] and is widely used to record from mouse photoreceptors with patch pipettes ([23] and subsequent studies from several labs). Since isolated retinas lack retinal pigment epithelium, we focused on the scotopic range. Thus, the metabotropic glutamate receptor (mGluR6) agonist 2-amino-4-phosphonobutyric acid (AP4) was used to block the b-wave, which, in these conditions, is mediated largely by rod (ON) bipolars [16].

Here, we present the results of our efforts, which consist of a compact and fully enclosed ex vivo ERG apparatus supporting four retinas (two ‘treated’ and two ‘controls’) in a stationary volume of only 2 mL/retina. Flash responses can be frequently obtained for over than one day, with only a slow progressive decrease in light sensitivity and stable kinetics. Examples of drug delivery and related data analysis are provided.

## 2. Results

### 2.1. Basic Description of the Apparatus and Setting up of an Experiment

Our approach to avoid superfusing the retinas was inspired by tissue cultures: provide adequate diffusive gaseous exchange by incubating them in wide and shallow wells, placed in a 95% O_2_/5% CO_2_ atmosphere saturated with water vapor. All relevant technical details of the system are given in the Materials and Methods. Two animals were dark adapted, their retinas isolated at room temperature and made to adhere to filter paper on their vitread side. They were then placed, filter paper down, at the bottom of two wells in a mobile insert block capable of simultaneously hosting four retinas (Figure 1a). The wells contained 4 mL of recording solution (2 mL/retina), which consisted of bicarbonate-buffered Ames’ medium with 40 µM AP4. The retinas were centered on the openings of channels leading to agar bridges (anodes) and were held in place magnetically by plastic rings that pressed on the supporting filter paper (Figure 1b). Each well also hosted an agar bridge (cathode/reference). The insert block was positioned on a partially submerged heated platform inside the recording chamber (Figure 1c). Platform temperature was automatically maintained at 35 or 37 °C, depending on the experiment. Injection tubing led to 1 mL syringes outside the chamber, one for each well. The anodes and cathodes/reference electrodes were inserted in their respective agar bridges. The chamber was sealed shut with a lid that had an LED light source (505 nm) attached on its internal face and flushed with a 95% O_2_/5% CO_2_ gaseous mixture. Sets of ‘flash families’ were delivered every 15 or 30 min and ERG responses recorded either in DC mode or with very slow high pass filtering (0.02 Hz cutoff) (Figure 1d).

### 2.2. Barium Is Not Required to Obtain Good Estimates of Light Sensitivity and Response Kinetics

Ex vivo ERG recordings enable the pharmacological isolation of the photoreceptor a-wave by removing the b-wave (here with the mGluR6 agonist AP4) and the slow glial wave with barium (Figure 1). Given the potential side effects of barium application [10], especially over the course of long experiments, we examined whether it was possible to obtain similar estimates of light sensitivity and response kinetics in AP4 as those in AP4 + BaCl_2_. As the barium sensitive PIII has a very slow time course, we focused on the initial phase of flash responses. Thus, as a proxy for light sensitivity, we selected the flash strength required for a half-maximal response (i_50_), where amplitudes were measured 90–130 ms after the flash (Figure 1, yellow band). Response kinetics were given by the time-to-peak at i_50_ (TTP@i_50_), estimated as the weighed average of the TTPs of the two flashes straddling i_50_. In a set of experiments at 35 °C, we injected 50 µM BaCl_2_ after 3–4 h of recordings. i_50_ and TTP@i_50_ were compared immediately before and 30–45 min after barium injection, when glial wave removal was largely complete. i_50_ before was 27.3 photons (ph)/µm^2^ (Hodges–Lehmann estimator; H-L) and after was 28.0 ph/µm^2^, with no significant difference detected (*p* = 0.93, *n* = 34, pW test). Moreover, the effect of barium was highly likely to be marginal, with a 99% confidence interval (CI_99%_) of the percentage change of −5.0% to 5.8% (CI of the H-L estimator of % change). TTP@i_50_ before was 143.0 ms and after was 129.3 ms, with a significant difference (*p* < 0.001; *n* = 34). The effect of barium on TTP was only moderate, with a CI_99%_ of the percentage change of −13.8% to −7.5%.

### 2.3. After an Initial Settling, Sensitivity and Kinetics Are Reasonably Stable over Several Hours

When the effects of a test solution are suspected to emerge over time, long recordings with stable conditions are required. We performed a large set of experiments with this apparatus by delivering flash families every 15 min with the retinas incubated in 40 µM AP4 (no barium). We compared the temporal evolution of flash responses at 35 °C (*n* = 43) and 37 °C (*n* = 13). Saturating amplitudes, measured in the range 90–130 ms after the flash, decreased over the course of several hours, rapidly initially followed by a slower progression (Figure 2a): from 0:45 to 4:45 h the Hodges–Lehmann estimator (H-L) of the instantaneous rate of change in amplitude was 17.9 µV/h @35 °C and 17.7 µV/h @37 °C. Such rate was significantly greater than zero in all time bins at both temperatures (*p* < 0.01; Wd-test), but in only 1 of 16 time bins was a difference detectable between them (MW-test). Light sensitivity decreased very slowly after an initial settling phase (Figure 2b): from 0:45 to 4:45 h, the H-L of the instantaneous rate of change in i_50_ was 2.32 ph·µm^−2^/h @35 °C and 2.07 ph·µm^−2^/h @37 °C. Such a rate was significant in all but two time bins at 35 °C (*p* < 0.01; W-test) and 9 of 16 bins at 37 °C (*p* < 0.05). A difference in rate between the two temperatures was detected only in a few initial bins (*p* < 0.01). Response kinetics was the most stable parameter (Figure 2c): from 0:45 to 4:45 h, the H-L of the instantaneous rate of change in TTP@i_50_ was 0.52 ms/h @35 °C and −2.9 ms/h @37 °C. Such a rate was significant in 1 of 16 bins @35 °C (*p* < 0.05; W-test) and 9 bins @37 °C (*p* < 0.01), with a difference between temperatures detectable in 5 bins (*p* < 0.05). Finally, we examined whether at 37 °C changes in i_50_ and TTP@i_50_ during the experiments were correlated with the time-dependent decay in amplitude. The distributions of rate of change (every 15 min) did not satisfy tests of normality so non-parametric Spearman’s correlations were used. Neither changes in amplitude vs. i_50_ (rho = 0.05; *p* = 0.35), nor did those in amplitude vs. TTP@i_50_ (rho = −0.03; *p* = 0.56) show a significant relationship. This suggested that whatever was causing the progressive rundown in amplitude did not seem to play a role in the slow decrease in light sensitivity.

### 2.4. Example Application 1 (Assessing Substance Effects over the Medium Term): Barium Chloride

While barium is routinely used to quickly eliminate the slow PIII wave in ex vivo ERG recordings [24], side effects on photoreceptor physiology cannot be ruled out [10], particularly when the exposure lasts for several hours. Here, we examined whether barium had any negative repercussions on response amplitude, light sensitivity and response kinetics, by adopting the following approach: (i) pairs of retinas were allowed to settle in 40 µM AP4 @37 °C for ~1 h; (ii) at conventional time 0:00 an aliquot of 1 mM BaCl_2_ in PBS was injected in one well to obtain a final concentration of 50 or 100 µM, and an equal volume of vehicle was injected in the other well; (iii) at 0:37.5 h barium was assumed to have fully blocked the glial wave and this was, thus, taken as the reference time; (iv) response amplitude (range 90–130 ms), i_50_ and TTP@i_50_ from the barium-treated retina were divided by those from the control, and the resulting ratios were normalized to that at the reference time (for details on analysis strategies see Methods). Within our sample, we did not detect significant effects of barium on any of the three parameters (beyond its initial rapid abolishment of the glial wave) up to 4.5 h from the injection (Figure 3). The CI_95%_ bands suggest that any actual effect would be very modest within this time frame.

### 2.5. Example Application 2 (mL-Scale Drug Testing): Pigment Regeneration after a Mild Bleach

As a demonstration of how this apparatus lends itself to testing very small drug quantities, we recorded pairs of retinas (*n* = 3) in 40 µM AP4 and 50 µM BaCl_2_ at 37 °C. Dim light (8270 ph·µm^−2^/s) was turned on for 10 min to induce a mild bleach, which was estimated to involve 3% of the rod pigment pool. Bleaching adaptation appeared as a reduction in light sensitivity by a factor of 1.9 (Figure 4a), very close to the value of 2.2 expected in mouse for a 3% bleach (Equation (1) with k = 40 in [25]). Moreover, response kinetics were accelerated (Figure 4b). Thereafter, one retina was rapidly and fully regenerated with 100 µM 9-cis-retinal, a concentration obtained by injection in the incubation/recording well of 57 µg retinoid for each retina (Figure 4a,b). Saturating flash responses displayed a marked increase in amplitude following regeneration (cf. C2 vs. C3 in Figure 4) consistent with a recovery in the circulating current and despite the typical rundown observed with this technique (Figure 2a).

### 2.6. Antibiotics Extend the Recordings to One Day and Beyond

The maximum duration of these recordings was dictated by the progressive decay of the flash response amplitudes (Figure 2a), as they eventually fell below the threshold at which sensitivity and kinetics could be reliably estimated (~5 µV for a saturating flash). In the conditions described thus far, this threshold was reached after 3–10 h depending on the retina. Among the many possible factors responsible for this decay, bacterial proliferation appeared both plausible and easy to test for: we included the broad spectrum antibiotic PenStrep (1% vol/vol) in the recording/incubation medium (37 °C; *n* = 16). In these recordings, flash families were delivered every 30 min. The effect of the antibiotic was dramatic, with the shortest recording lasting more than 13 h (Figure 5a). In fact, several retinas continued to show flash responses beyond 36 h of incubation. This extreme duration was associated with a smaller rate of change in amplitude in PenStrep compared to untreated retinas: statistical significance was detected from 2:30 h onwards (*p* < 0.05 in 4 of 15 bins and *p* < 0.01 in 8 of 15 bins of 30 min each; Wd-test). In contrast, the rates of change in i_50_ with and without PenStrep were not significantly different in 12 of 15 bins of 30 min after 2:30 h (MW-test). From 8:00 to 42:30 h, the H-L of the instantaneous rate of change in i_50_ in PenStrep was 1.19 ph·µm^−2^/h, amounting to a slow drift (Figure 5b). Additionally, response kinetics did not seem to be affected by antibiotic, with TTP@i_50_ not significantly different (MW-test) in 13 of 15 bins of 30 min (after 2:30 h). From 8:00 to 42:30 h, the H-L of the instantaneous rate of change in TTP@i_50_ with PenStrep was 0.076 ms/h, indicative of long-term stability (Figure 5c). Having removed fast-progressing bacterial contamination with PenStrep, we examined whether the slow decrease in i_50_ was correlated with the residual decay in response amplitude. However, changes in amplitude vs. i_50_ again did not show a significant relationship (rho = 0.05; *p* = 0.15).

Over the course of this latter batch of very long experiments, the osmolality of the incubation medium had a marked tendency to increase, in all likelihood due to net evaporation from the wells. The initial value of 280 mOsm/kg of Ames’ medium was found to reach 300–330 mOsm/kg at the end of the recordings and, occasionally, much above. Surprisingly, even extreme shifts in osmolality did not compromise phototransduction: two retinas were still responding after 37 h (Figure 5, black arrows; Figure 6) when recordings were interrupted and osmolality in their shared well measured at 396 mOsm/kg! Perhaps the photoreceptors were able to adapt to such hypertonic conditions due to how slowly they developed. No systematic attempt was made to correct this upward drift by distilled water injection.

## 3. Discussion

We showed that isolated mouse retinas, maintained at 37 °C in as little as 2 mL of Ames’ medium, are able to express scotopic a-wave responses that are relatively stable in sensitivity, kinetics and trajectory over the course of more than one day (Figure 5 and Figure 6). This occurs sufficiently frequently to make statistically meaningful medium-term testing of drugs feasible. Normalization to pre-treatment values and division of ‘treated’ retina over the ‘control’ reduce variability and, therefore, also the necessary sample sizes. The small incubation volumes mean that expensive molecules or nano-device suspensions can be tested at high concentrations, again implying fewer animals. Preliminary screening of drugs in such an ex vivo retina system avoids all issues and uncertainties related to systemic (i.e., vascular) or intraocular transport and delivery. It is, thus, agnostic in terms of the nature of the formulation being tested, irrespective of its molecular weight (including recombinant proteins [4]) and hydrophobicity (e.g., 9-cis-retinal; Figure 4), spanning from simple solutions to suspensions, emulsions and nano-carriers such as liposomes [4] and exosomes [26]. Lastly, concurrent recordings from four retinas double the experimental throughput.

Light responses from isolated mammalian retinas kept at body temperature for long periods were sporadically reported before. One example is Figure 3 in Ames and Nesbett’s original paper describing their optimized medium [21], where the rabbit retinal ganglion cells discharged upon light stimulation light even after 51 h. Much later, some changes in transretinal potential were detected when stimulating with bright light a rat retina cultured for 4 days (Figure 4 in [27]; 26 mL/retina substituted daily). Recently, ERG-like responses (miERG) were recorded via a multi-electrode array from organotypically cultured adult mouse retinas after one and two days (Figure 3 in [28]); these, however, required flash strengths at least three orders of magnitude higher than what we used in the present study. Moreover, each retina explant could be recorded at only a single time point. Given this background, we were surprised to find that just 2 mL of incubation medium sufficed for maintaining rod phototransduction in a seemingly physiological state for at least a day and a half.

Pushing the envelope in terms of how long the mouse retina can be kept physiologically responsive ex vivo requires identifying, iteratively and often through trial-and-error, the factors behind the dominant time constants of the response parameters’ drift or decay. Here, starting from the initial performance of our novel system, we identified bacterial proliferation as one crucial factor in amplitude decay during the first 24 h of incubation. Inclusion of a broad spectrum antibiotic in the basic medium greatly improved long term retina viability. Due to the intentional absence of open loop superfusion in this system and not having enforced sterile conditions during dissection and incubation, this was hardly surprising. Our present data complement studies that examined the acute retinal toxicity of several antibiotics with ex vivo ERG and found the a-wave to be unaffected [29,30]. It must be noted that organotypic retina cultures frequently include antibiotics in the media, in addition to enforcing sterile dissection and incubation conditions [19,27,31,32]. The use of PenStrep greatly extended our experiments, but further improvements will require identifying the causes of the residual amplitude decay. A similar phenomenon, albeit occurring on much shorter time scales, was investigated in the context of the b-wave, which is rather more labile than the a-wave [8]. This was linked to a depletion of glutamate at the rod–rod bipolar synapse [9] or to insufficient tissue superfusion and associated physicochemical disequilibria [6,11,14,18,22]. In the present case, the more obvious factors that will need to be excluded are: (i) nutrient (or O_2_) depletion or metabolite (or CO_2_) accumulation in the 2 mL available to each retina; (ii) pH drifting out of the physiological range; (iii) proliferation of antibiotic-resistant bacteria or fungi; (iv) leaching of silver ions or other contaminants [21]. Arguing against these four candidates are our observations of retinas incubated in the same well (in PenStrep) that showed strikingly different survival times. Another testable hypothesis is slow detachment from the filter paper. Among the more exotic possibilities is a reduction in the resistance of the extracellular space, as was postulated for the b-wave in early studies [8,21]. We can instead exclude a major role for the upward drift in osmolality, since we observed the relatively early disappearance of the a-wave in retinas incubated in only slightly hypertonic medium, while exceptionally long-lasting retinas were still responding in 396 mOsm/kg medium (Figure 6). In spite of this, future improvements to our system will need to address the underlying net water evaporation from the incubation wells.

With regard to light sensitivity, its slow progressive reduction was not significantly correlated to the decay in amplitude, implying that the decrease in the circulating current associated with this loss of sensitivity (cf. amplitudes in Figure 4(c1–c3)) played a marginal role in the latter process. Unexpectedly, this upwards drift in i_50_ could not be explained by bleaching adaptation caused by the repeated delivery of flash families, since each such sequence of flashes was estimated to bleach only 0.005% of the rhodopsin complement. This would be expected to cause an increase in i_50_ of not more than 0.2% (Equation (1) with k = 40 in [25]), while the observed increase was in the range of 1–3%. In the future, it will be worthwhile to determine the effects of 9-cis-retinal injection after many hours of incubation, as these might clarify whether the issue is ‘simply’ one of photopigment bleaching, or whether more fundamental changes to the phototransductive cascade are involved.

Lastly, response kinetics were surprisingly resilient to long term incubation, an observation that contrasts starkly with their instability in single cell recordings [33] and further supports the argument that the ex vivo ERG provides estimates of unperturbed kinetics [34].

The potassium channel blocker barium is routinely used in ex vivo ERG recordings to abolish the slow PIII wave mediated by Müller glia (Figure 1d and [24]). Given the importance of these cells in retinal homeostasis [35] and considering that barium has other more subtle effects directly on photoreceptors [10], we were concerned that, in long experiments, it might become cytotoxic. However, we did not identify any overt noxious effects barium over several hours of retinal exposure (Figure 3). Future studies will need to examine even longer exposures, in parallel with increases in the length of ex vivo ERG recordings. In any case, here we have shown that when in doubt over whether this blocker may interfere with one’s own application, it can be dispensed with it while still obtaining good estimates of i_50_ and TTP@i_50_. In the short term, barium did not affect scotopic sensitivity and its moderate hastening effect on kinetics was likely to be, at least in part, an epiphenomenon of the abolishment of the slow glial wave.

Looking ahead, the well established techniques of organotypic culture of adult mouse retinas provide a convenient benchmark for what might be achievable with ex vivo ERG. When retinas are cultured without pigment epithelium (RPE), photoreceptor apoptosis is first detected after 3–4 days and rapidly increased thereafter, leading to a significant thinning of the outer nuclear layer from 5 to 6 days onwards [32]. However, these events are preceded in the first 4 days by severe alterations in the morphology of rod outer and inner segments. It is, therefore, possible that with this approach, physiological a-wave light responses will not be obtainable beyond 2–3 days. An alternative method, in which the RPE is co-cultured with the retina, is promoted as offering improved long-term viability ([36] and references therein). While its use was mostly limited to neonatal or early postnatal mice, it may constitute the key to extending chronic ex vivo ERG recordings beyond a few days.

Major efforts are under way to establish good in vitro and ex vivo alternatives to the in vivo retina, one major aim being faster and cheaper development of therapies for ocular diseases. In vitro technologies are at various levels of maturity, such as primary retinal cell cultures, retina-derived cell lines, retinal organoids, retinas-on-a-chip and bioprinted retinas [20,37]. Important advances were also made in the long term organotypic culture of retinal explants [20,32,36]. All these models offer direct pharmacological access and, potentially, longitudinal monitoring of effects, but lack the expression of the physiological light responses generated by in vivo or acutely isolated retinas. Our study contributes to bridging the gap by extending ex vivo ERG recordings of the a-wave from a few hours to more than one day, while limiting the incubation medium to a mere 2 mL. It must be noted that, as for all of the aforementioned ‘reduced’ model systems, any discoveries made with this apparatus will eventually need to be validated in the living animal.

## 4. Materials and Methods

### 4.1. Dissection and Handling

Adult (>P30) C57Bl/6J mice were dark adapted, sacrificed by i.p. injection of urethane (20% w/vol in saline) and their retinas immediately extracted in basic medium, without the RPE, through a corneal incision. One modification from previous work, both ours [23,38] and of other labs, was that throughout extraction and preparation the medium not cooled but left at room temperature. Retinas were cleansed from vitreous humor and made to adhere to filter paper (5 µm, SMWP02500; Merck, Darmstadt, Germany) on their vitread side, by gentle transmural suction provided by a peristaltic pump. Excess filter paper was trimmed with scissors and this retina-paper sandwich transferred with curved tweezers into a well of a mobile insert block (Figure 1a). The two eyes from an animal were moved to separate wells.

### 4.2. Insert Block

This consisted of nylon block (74 × 52 mm, height 16 mm) milled with two oval wells (22 × 40 mm, height 10 mm), each hosting two retinas. These were centered on holes (diam. 1.4 mm; Figure 1a, ch) which led, via channels, to insert points for the anodes (diam. 2 mm; Figure 1a, an). Each well had a slot drilled on its side for the shared cathode/reference electrode (diam. 2 mm; Figure 1a, ca). Each well and its electrode channels were filled with 4 mL of basic medium. The paper supporting the retinas was held in place (Figure 1b) by a polycarbonate ring (Figure 1a, ri) and small nickel-coated neodymium magnets (diam. 2 mm, Magfine Corporation, Sendai, Japan; Figure 1a, mag), the latter covered with a thin layer of glue or silicone grease to minimize the risk of metal ions release. A variant was also developed in which the retinas were not attached to the filter paper but were held in place by precisely sized polycarbonate cups (Figure 1b). Glass pipettes filled with 1.2% w/vol agarose [13] were inserted in their respective positions (Figure 1a, an, ca) to act as agar bridges and minimize the release of silver ions in the recording solution.

### 4.3. Enclosing Chamber

The chamber (Figure 1c) consisted of a PMMA enclosure (230 × 190 mm, height 120 mm; wall thickness 10 mm) and PMMA cover lid (230 × 190 mm; wall thickness 20 mm), covered with adhesive neoprene insulation. The chamber housed at its bottom a three-layer sandwich: anodized aluminum platform (194 × 164 mm; thickness 10 mm), silicone resistive heating pad (mod. 3316100, Thermo Technologies/Conrad, Hirschau, Germany; 150 × 180 mm, ~0.7 Ohm) and anodized aluminum base (194 × 164 mm; thickness 5 mm); the two aluminum plates were screwed together so as to lightly squeeze the heating pad; the power cable was routed out via a hole in the side of the chamber (gasket). The temperature of the aluminum platform was monitored with an IC probe connected to a custom-build temperature controller [39]. Since this was a bipolar controller designed for driving thermoelectric (i.e., Peltier) devices and not resistive loads, a power diode was inserted along the cable to the heating pad. Descaled and lightly deionized water (reverse osmosis) was poured in the chamber until the platform was entirely covered. Grounding connections for the plates and water bath were ensured by gold-coated pellets, sockets and pins (Mill-Max Mfg. Corp., Oyster Bay, NY, USA) embedded in the plates’ aluminum core using electrically conductive adhesive (#8330S; MG Chemicals, Burlington, ON, Canada). The insert block was placed on the platform and held in position by 3 pairs of embedded magnets. Chlorided silver wire electrodes (0.25 mm), one anode for each retina plus two interconnected cathodes/references, were dipped in the agar bridges and fixed in position by magnetic supports. For each well, one end of a PTFE injection tubing (ID = 0.5 mm; OD = 1 mm) was inserted in a silicone holder in a receptacle (Figure 1a, t), while the other end exited the chamber via a small hole and reached a 1 mL syringe.

The light source (505 nm LED) and ND filters (50 × 50 mm) were housed in a 3D printed sealed box attached centrally to the chamber cover lid. The inferior surface of the box facing the preparation consisted of indium tin oxide coated glass (50 × 50 mm; #1310; Adafruit, New York, NY, USA) through which sufficient DC current was constantly passed, side to side, to keep its temperature above the dew point, thereby preventing condensation on the glass. Also attached to the cover lid, below the LED box, was a thin transparent polycarbonate sheet inclined at an angle, whose aim was to prevent water condensation on the lid from dripping into the underlying wells. This was found to be important for recordings lasting > 1 day. The cover lid was sealed shut (perimetric gasket) and the inner air space of the chamber flushed with 95% O_2_/5% CO_2_ gas mixture for a few minutes; a minimal flow rate (~1 bubble/s) was, thereafter, maintained throughout the experiment.

### 4.4. Solutions

Basic medium, used throughout dissection and recordings, consisted of bicarbonate-buffered Ames’ medium (A1420, Merck, Darmstadt, Germany) supplemented with 40 µM of the mGluR6 agonist AP4 (0101, Tocris, Bristol, UK). Water for solutions was first descaled, it was then passed through reverse osmosis and, finally, single distilled (model 500100; Kontes Glass, Vineland, NJ, USA). The following substances were also employed in specific experiments. BaCl_2_ was prepared as a 1 mM stock solution in PBS to avoid the formation of sulfate precipitate and added or injected in the recording chamber at a dilution of 1:20 (50 µM) or 1:10 (100 µM). 9-cis-retinal (R5754; Merck, Darmstadt, Germany) was prepared as a 100 mM stock solution in ethanol and frozen in aliquots; on the day of the experiment a 4 µL aliquot was diluted in 200 µL modified medium and injected in the well to obtain a final concentration of 100 μM; in these experiments, the recording and incubation medium was modified by supplementing basic medium with 1% w/vol fatty acid-free BSA (A8806; Merck, Darmstadt, Germany) as a retinoid solubilizing agent [40]. Where used, PenStrep (SV30010; HyClone, Logan, UT, USA) was diluted on the day of the experiment at 1% vol/vol. Media osmolality was measured with a Type 7 osmometer (Löser Messtechnik, Berlin, Germany).

### 4.5. Electrophysiological Recordings and Automated Parameter Extraction

The potential differences between anodes and cathode/reference electrodes were filtered in the range DC-100 Hz or 0.02–100 Hz and amplified by 5000 using a custom-built amplifier, digitized at 5 kHz with a Digidata 1322B and recorded with pClamp 9 (Axon Instruments, Union City, CA, USA). Electrophysiological records were batch processed in Axograph X 1.7.6 using custom scripts. Briefly, for each flash family and channel: (i) flashes of the same strength were averaged; (ii) any sloping baseline was removed by subtracting a linear fit of the data in the 1 s before the flash; (iii) response amplitudes were taken as the average in the range 90–130 ms after the flash (Figure 1d, yellow band); (iv) i_50_ was determined from the Hill fit to a plot of these amplitudes vs. flash strengths; (v) TTP@i_50_ was estimated, on 10 Hz Gaussian filtered records, as the weighted average of the TTPs of the two flashes straddling i_50_. Automation of this entire pipeline meant that an entire experiment consisting of tens of recordings could be processed in a matter of minutes.

### 4.6. Data Processing Pipeline

When testing the effects of a drug or formulation, one retina from an animal was ‘treated’ and the other used as its paired ‘control’. Treatment and control wells were alternated from experiment to experiment to cancel out any subtle environmental biases. We assumed that the two retinas, being from the same animal, behaved identically except during the initial stabilization phase and, irrespective of time, for a fixed scaling factor (Figure 7a). The latter assumption seemed reasonable given the inevitable differences in retina dissection and handling, as well as shape, position and orientation of the retina-filter paper sandwiches in the wells. For instance, transretinal shunt resistances determined the attenuation of the response amplitudes recorded at the electrodes. Similarly, differences in pigment bleaching during the initial dissection and of the orientation of outer segments during the recordings, affected light sensitivity.

Two rounds of normalization were applied to the response amplitude, i_50_ and TTP@i_50_ pairs of time series extracted as described in the previous paragraph. First, ‘treated’ and ‘control’ retinas were normalized over their respective pre-treatment values (Figure 7b). Second, any long-term parameter drifts common to both retinas were removed by time-wise division of ‘treated’ over ‘control’ (Figure 7c). We were, thus, left with a single time series for each parameter, whereby values above unity signaled an increase due to the drug and vice versa (for an example, see Figure 3, red lines).

### 4.7. Statistics

Statistical analyses were performed with the open source software JASP 0.16 (jasp-stats.org; RRID:SCR_015823). Instantaneous rates of change were calculated as the Hodges–Lehmann estimator [41] of change from one flash family to the next (for all retinas at the same 15 or 30 min bin) and reported as hourly rates. Visual representations of the population effect of a tested drug were given by the Hodges–Lehmann estimator and its confidence interval (Figure 3, thick lines and colored areas). Accordingly, statistical significance was estimated by the following non parametric tests: paired Wilcoxon signed-rank (pW-test), one sample Wilcoxon directional signed-rank (Wd-test), one sample Wilcoxon signed-rank (W-test), Mann–Whitney (MW-test).

## Figures and Tables

**Figure 1 ijms-24-11346-f001:**
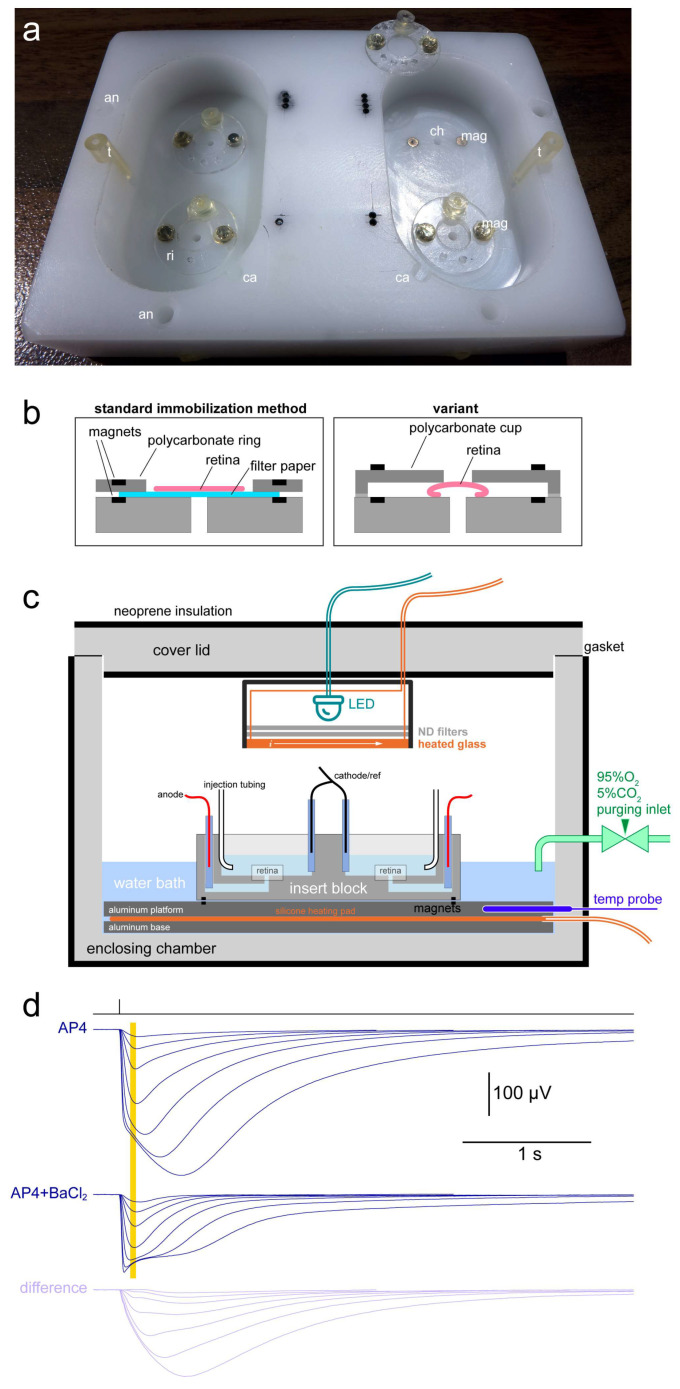
The ex vivo ERG apparatus. (**a**) Insert block for simultaneously recording 4 retinas. Each well (control or treatment) hosted two retinas and was filled with 4 mL of medium. an: anode insertion point; ca: cathode/reference insertion point; ch: channel leading to anode; ri: ring for holding retina in position; mag: magnet; t: injection tubing holder. (**b**) In the standard method of immobilization (**left**) the retina was supported by filter paper, which in turn was held down by a polycarbonate ring and tiny magnets. An alternative method (**right**) dispensed with the filter paper by slightly compressing the retina with a polycarbonate cup; while being mechanically less stable, this method should avoid any capture of the drug formulation (e.g., nanoparticles) by the porous filter paper. (**c**) Labelled schematic of the enclosing chamber and insert block. (**d**) Sample flash families obtained at 37 °C from a retina in Ames’ + 40 µM AP4 (**top**) and after injection of BaCl_2_ in the well to achieve a final concentration of 100 µM (**center**). Also shown is the difference between the two sets of responses (**bottom**). Each trace was the average of several flash responses, with an entire flash family delivered every 15 or 30 min. Flash strengths (ph/µm^2^)|no. of repetitions: 3.98|12, 8.27|10, 18.9|8, 50.5|6, 151|6, 510|4, 1660|3. The yellow band shows the range in which response amplitudes were measured (90–130 ms after the flash) and used to fit dose–response curves and derive light sensitivity (i_50_).

**Figure 2 ijms-24-11346-f002:**
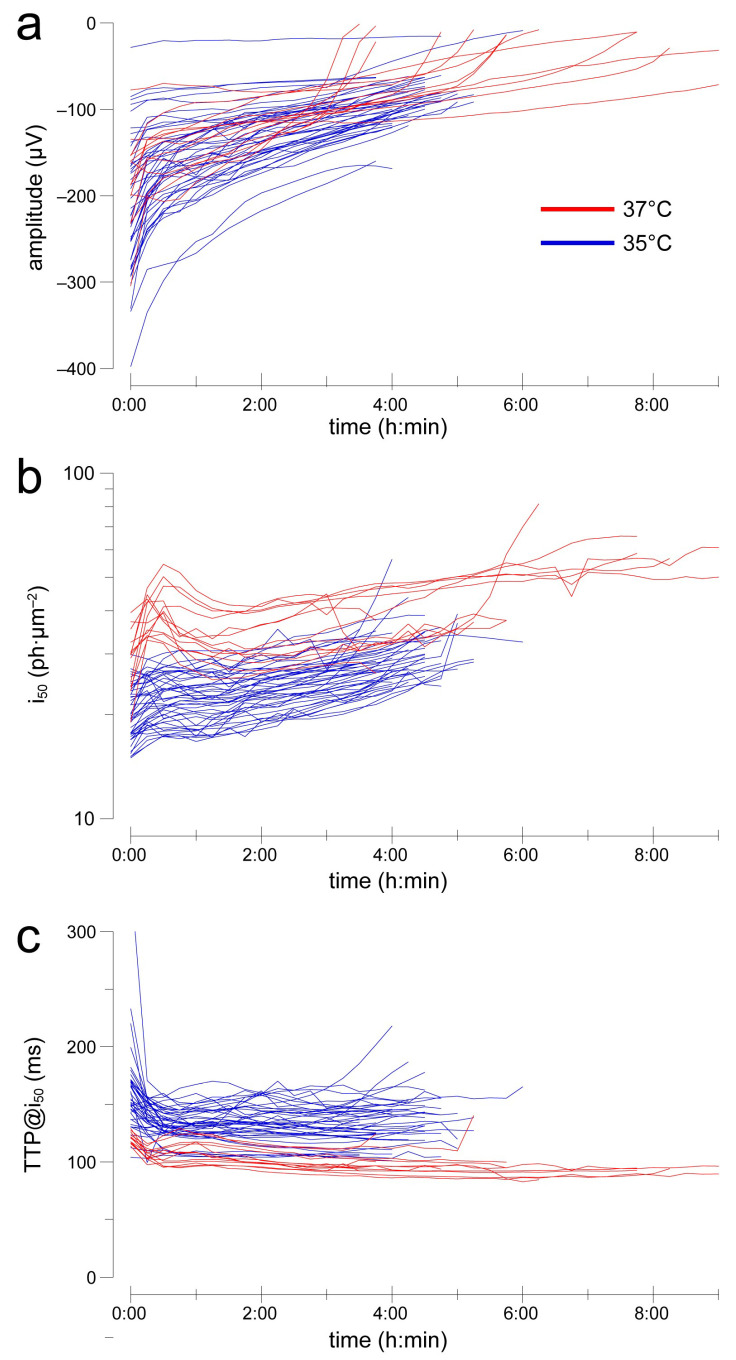
Evolution of flash response amplitude (**a**), sensitivity (**b**) and kinetics (**c**) during recordings in 40 µM AP4 at 35 °C (blue) or 37 °C (red). Each line shows the data from an individual retina. Note that most of the recordings at 35 °C were intentionally capped at 4–5 h duration.

**Figure 3 ijms-24-11346-f003:**
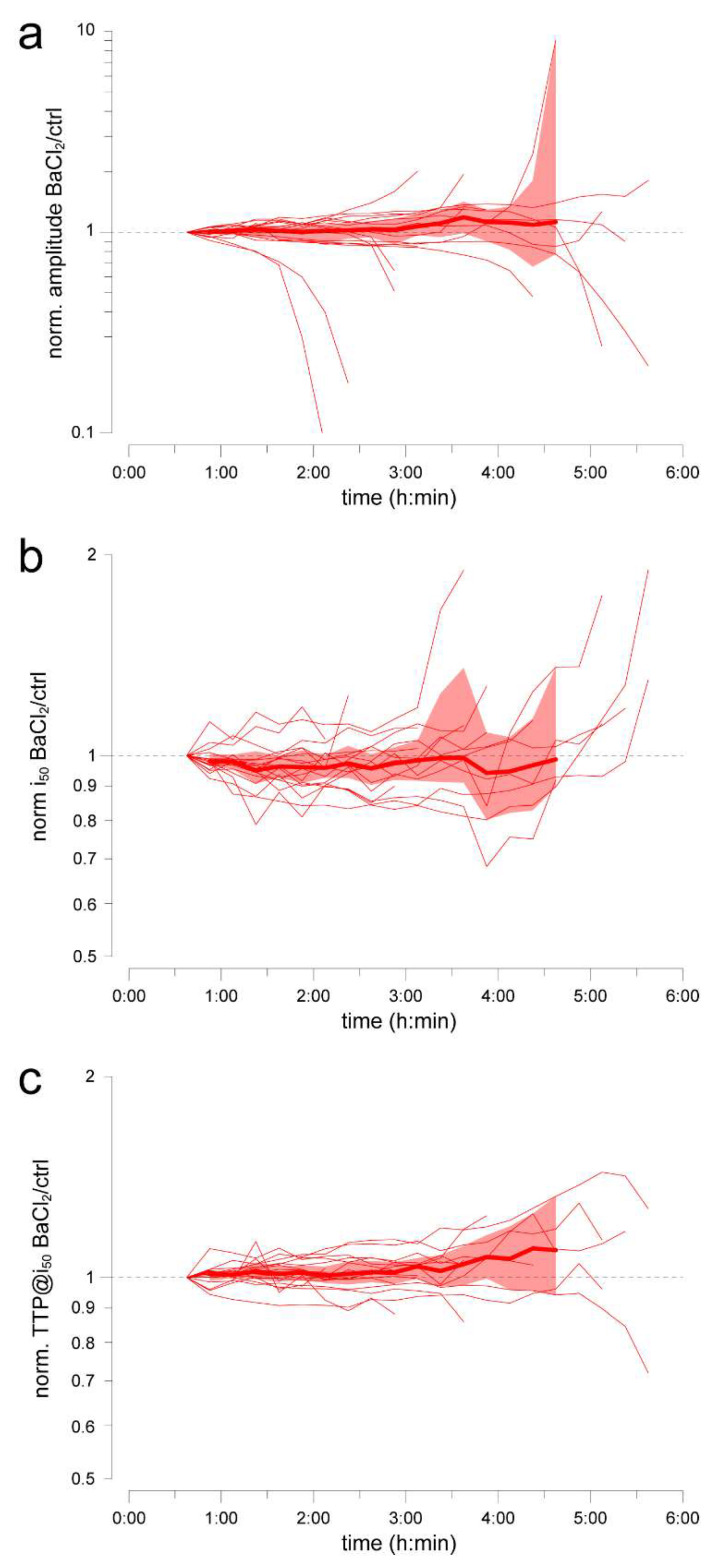
Relative evolution of flash response amplitude (**a**), sensitivity (**b**) and kinetics (**c**) after administration of 50–100 µM BaCl_2_. Recordings were carried out in 40 µM AP4 at 37 °C. At time 0:00, BaCl_2_ was injected in one well and vehicle (PBS) in the other. Thin lines: evolution of the normalized ratio of individual retina pairs; thick lines: Hodges–Lehmann estimators; colored areas: CI_95%_.

**Figure 4 ijms-24-11346-f004:**
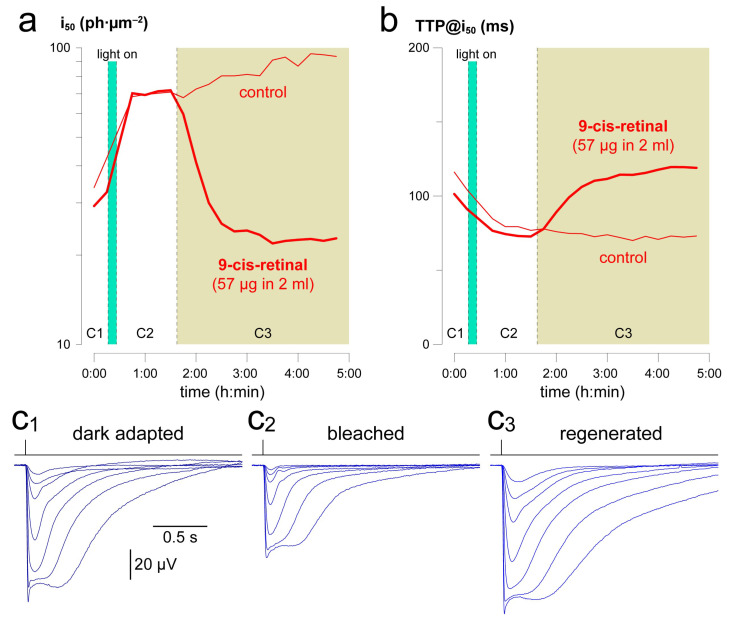
Light-evoked pigment bleaching followed by full regeneration with a small amount of 9-cis-retinal. Time evolution of i_50_ (**a**) and TTP@i_50_ (**b**) after a 10 min light exposure and after injection of 57 µg 9-cis-retinal or vehicle. Thick line: regenerated retina; thin line: control retina; green bar: light exposure; sand area: retinoid/vehicle present in the wells. Average flash responses at the beginning of the experiment (**c1**), after pigment bleaching (**c2**) and after regeneration (**c3**). Recordings were carried out in 40 µM AP4 and 50 µM BaCl_2_ at 37 °C.

**Figure 5 ijms-24-11346-f005:**
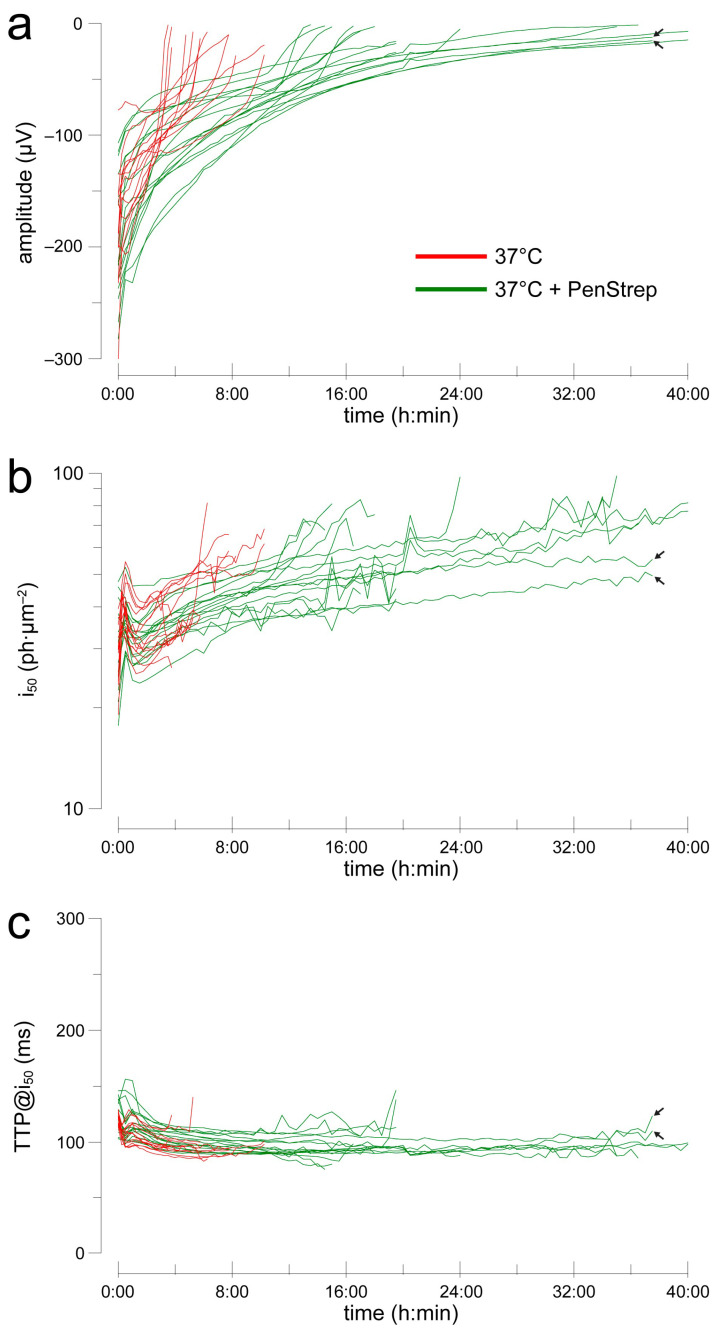
Evolution of flash response amplitude (**a**), sensitivity (**b**) and kinetics (**c**) during recordings at 37 °C in 40 µM AP4 and 1% PenStrep (green), compared to those without antibiotic (red). Each line shows the data from an individual retina. Arrows point to the retinas shown in Figure 6.

**Figure 6 ijms-24-11346-f006:**
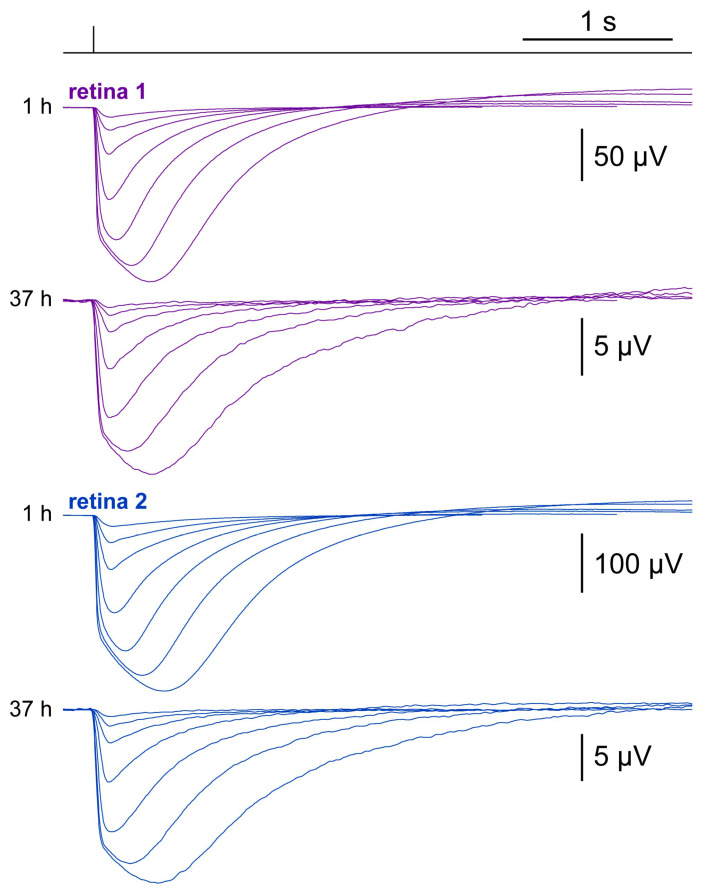
Average flash responses of two retinas incubated at 37 °C in 1% PenStrep, showing minimal changes in waveform even after 37 h and despite a dramatic increase in medium osmolality (see text).

**Figure 7 ijms-24-11346-f007:**
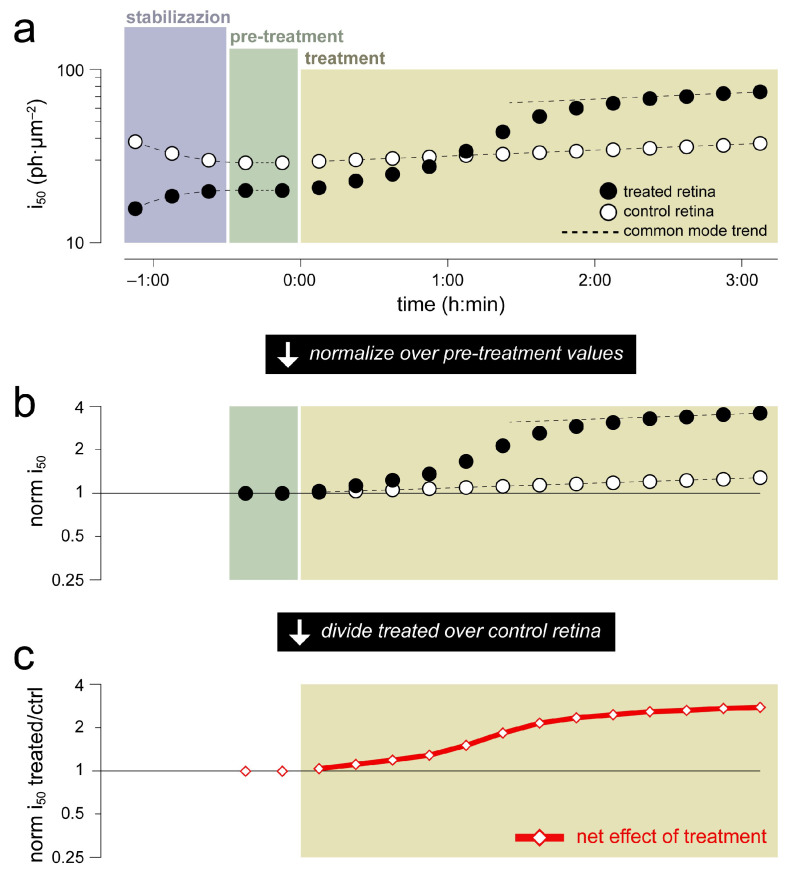
Processing pipeline of the ex vivo ERG data (i_50_ in this example, but the same is applicable to other parameters) obtained in a hypothetical experiment where a retina is incubated with a test drug, while the other retina is given vehicle (i.e., control). (**a**) We assume that the two retinas, being from the same animal, behave identically except for: (i) an initial stabilization phase due to differences in their isolation and manipulation; (ii) a constant scaling factor (in the case of i_50_ this would be due to the particular retinal geometry in the wells). (**b**) We normalize the raw parameter values by their respective pre-treatment levels. (**c**) We remove any common mode trends by dividing the normalized values of the treated retina by those of the control on, obtaining a single time series that shows the net effect of the treatment (in the example give here a threefold decrease in light sensitivity).

## Data Availability

Any of the data presented in this study are available on request from the corresponding author. The data are not publicly available due their considerable size and heterogeneity.

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
