# Peer review of "An Ex Vivo Electroretinographic Apparatus for the mL-Scale Testing of Drugs to One Day and Beyond"

_ijms, 2023, doi:10.3390/ijms241411346_

Round 1

Reviewer 1 Report

The current manuscript is an interesting study on the creation of a new ex vivo ERG apparatus for the study of ocularly delivered drugs. Its methodology appears to be quite sound, and the developed method could be quite useful for future studies in ocular therapeutics. Nevertheless, some small alterations should be made before acceptance for publication:

- In the introduction section, a reference is missing for the sentence “Furthermore, the ex vivo configuration allows the accurate and reproducible administration of drugs, receptor or ion channel blockers, modulators, small molecules as well as full proteins, without the hindrance of the ocular or blood-brain barriers.”;

- Figures should appear right after the paragraph in which they are first mentioned in the text, hence authors should reposition them;

- Figure quality should be improved, consider restructuring the graphs/images to make them individually bigger, so that words are more easily read;

- Abbreviations should always be defined in the text before being used, authors should check and correct this (for example “ERG”, which is defined in the abstract, but the abstract “does not count”);

- A part of Figure 7 caption is not visible, this should be corrected;

- Authors should discuss whether the developed method might be applicable for all types of drugs (low MW vs high MW, hydrophilic vs hydrophobic, etc.), and if all types of formulations could be evaluated (solutions, suspensions, emulsions, nanosystems, etc.).

Author Response

REVIEWER 1

The current manuscript is an interesting study on the creation of a new ex vivo ERG apparatus for the study of ocularly delivered drugs. Its methodology appears to be quite sound, and the developed method could be quite useful for future studies in ocular therapeutics. Nevertheless, some small alterations should be made before acceptance for publication:

- In the introduction section, a reference is missing for the sentence “Furthermore, the ex vivo configuration allows the accurate and reproducible administration of drugs, receptor or ion channel blockers, modulators, small molecules as well as full proteins, without the hindrance of the ocular or blood-brain barriers.”;

RESPONSE: Three citations have now been added. Two are previous ex vivo ERG studies that argue the advantages of this technique over its in vivo counterpart when testing drugs. The third is a published preprint by the present authors and colleagues, currently under review at the journal ‘Cellular and Molecular Life Sciences’, which studies the effects of ex vivo retinal incubation of recombinant proteins from both the morphological and ex vivo ERG points of view.

- Figures should appear right after the paragraph in which they are first mentioned in the text, hence authors should reposition them;

- Figure quality should be improved, consider restructuring the graphs/images to make them individually bigger, so that words are more easily read;

RESPONSE: We’ve greatly increased figure sizes and moved them to align with their respective paragraphs. Please note that we’ve significantly struggled with the journal formatting and styles in MS Word for Mac. Despite our best efforts some figures and legends could appear misaligned on the reviewer’s Word. We thus suggest to inspect the pdf version, which should guarantee to appear as we intended.

- Abbreviations should always be defined in the text before being used, authors should check and correct this (for example “ERG”, which is defined in the abstract, but the abstract “does not count”);

RESPONSE: We have now defined ERG again in the Introduction. Also in the Introduction “mGluR6 agonist AP4” has been replaced with “metabotropic glutamate receptor (mGluR6) agonist 2-amino-4-phosphonobutyric acid (AP4)”. In the Results: “ph/µm2” has been replaced with “photons (ph)/µm2”, “CI99%” with “99% confidence interval (CI99%)”, “(Hodges-Lehmann estimator) with “(Hodges-Lehmann estimator; H-L)”

- A part of Figure 7 caption is not visible, this should be corrected;

RESPONSE: We apologize to the reviewer for the misformatting. This issue was inadvertently introduced by the automatic file conversion system of the journal after our initial submission. We immediately informed the editorial office and were able to replace the file after a couple of days. We suspect that the reviewer probably dowloaded the first version of our manuscript.

- Authors should discuss whether the developed method might be applicable for all types of drugs (low MW vs high MW, hydrophilic vs hydrophobic, etc.), and if all types of formulations could be evaluated (solutions, suspensions, emulsions, nanosystems, etc.).

RESPONSE: We thank the reviewer for this suggestion which led to the addition of the following paragraph at the beginning of the Introduction: “Preliminary screening of drugs in such an ex vivo retina system avoids all issues and uncertainties related to systemic (i.e. vascular) or intraocular transport and delivery. It is thus agnostic in terms of the nature of the formulation being tested, irrespective of its molecular weight (including recombinant proteins [4]) and hydrophobicity (e.g. 9-cis-retinal; Fig. 4), spanning from simple solutions to suspensions, emulsions and nano-carriers such as liposomes [4] and exosomes [26].”

Reviewer 2 Report

The manuscripts presents a prototype ex-vivo ERG apparatus for drug evaluation. The data is well presented and aim of the study is interesting. However, following comments may be considered to improve the manuscript;

1. Title of the manuscript should not contain jargons.

2. Abstract should contain key findings of the research. 

3. In Discussion, some references about bioprinted retinal model may be added to draw a comparison.

4. How about retinal blood supply effect drug response. The apparatus may consider about this or give a brief explanation about that.

5. How about validation step of the apparatus compared with real-time conditions? Some steps may be added in the conclusion sections as future prospects.

Author Response

REVIEWER 2

The manuscripts presents a prototype ex-vivo ERG apparatus for drug evaluation. The data is well presented and aim of the study is interesting. However, following comments may be considered to improve the manuscript;

1. Title of the manuscript should not contain jargons.

RESPONSE: We assume that the reviewer is referring to the use of the acronym “ERG” in the title. We agree with this comment and have thus reformulated the title to: “An ex vivo electroretinographic apparatus for the mL-scale testing of drugs to one day and beyond”

2. Abstract should contain key findings of the research.

RESPONSE: We thank the reviewer for pointing this out and are pleased to report that the Abstract has now been extensively re-written. All changes are highlighted in red in the revised manuscript.

3. In Discussion, some references about bioprinted retinal model may be added to draw a comparison.

RESPONSE: We have taken up and expanded on this helpful suggestion by placing our study in the context of the broad spectrum of in vitro technologies being developed to study therapies for retinal diseases. Among them, of course, retinal bioprinting. This has led to the addition of a new ‘Conclusions’ section, which includes the following text: “Major efforts are under way to establish good in vitro and ex vivo alternatives to the in vivo retina, one major aim being faster and cheaper development of therapies for ocular diseases. At various levels of maturity are in vitro technologies such as primary retinal cell cultures, retina-derived cell lines, retinal organoids, retinas-on-a-chip and bioprinted retinas [20,37]. Important advances have also been made in the long term organotypic culture of retinal explants [20, 32, 36]. All these models offer direct pharmacological access and, potentially, longitudinal monitoring of effects, but lack the expression of the physiological light responses generated by in vivo or acutely isolated retinas. Our study contributes to bridging the gap by extending ex vivo ERG recordings of the a-wave from a few hours to more than one day, while limiting the incubation medium to a mere 2 ml.”

4. How about retinal blood supply effect drug response. The apparatus may consider about this or give a brief explanation about that.

RESPONSE: In a newly added paragraph at the beginning of the Discussion section we stress the point that testing drugs in the ex vivo incubated retina, bypasses all issues related to systemic (i.e. vascular) administration and is thus appropriate in the preliminary screening phase. The text reads as follows: “Preliminary screening of drugs in such an ex vivo retina system avoids all issues and uncertainties related to systemic (i.e. vascular) or intraocular transport and delivery. It is thus agnostic in terms of the nature of the formulation being tested, irrespective of its molecular weight (including recombinant proteins [4]) and hydrophobicity (e.g. 9-cis-retinal; Fig. 4), spanning from simple solutions to suspensions, emulsions and nano-carriers such as liposomes [4] and exosomes [26].”. We hope that this adequately addresses the suggestion made by the reviewer.

5. How about validation step of the apparatus compared with real-time conditions? Some steps may be added in the conclusion sections as future prospects.

RESPONSE: We have added the following final sentence to the Conclusions section: “It must be noted that, as for all of the aforementioned 'reduced' model systems, any discoveries made with this apparatus will eventually need to be validated in the living animal.”